# An Integrated Fuzzy Model for Selecting Resilient Suppliers in Electronics Industry of Iran

Hamzeh Aghababayi  and Mohsen Shafiei Nikabadi *

Industrial Management Department, Semnan University, Semnan 35131-19111, Iran;
hamzehaghababayi@semnan.ac.ir
* Correspondence: shafiei@semnan.ac.ir; Tel.: +98-912-540-4808

**Abstract:** Selecting appropriate and resilient suppliers is an important issue in supply chain management (SCM) literature. Making an effective decision on this issue can decrease external risks and disruptions, purchase costs, and delay times and also guarantees business continuity in the event of disruptions and, consequently, increases company competitiveness and customer satisfaction. This paper aims to provide a model based on identifying and investigating related criteria to evaluate suppliers' resilience and select the most resilient suppliers in Iran's electronic industry. To this purpose, the screening technique, the best-worst methodology (BWM), and goal programming (GP) have been applied in the fuzzy environment. The proposed model has been implemented and demonstrated by a case study of the electronic industry, as a real-life example. The results show that agility (0.227), compatibility (0.153), and vulnerability (0.102) are the most important factors for a resilient supplier.

**Keywords:** supply chain; supplier; resilience; best-worst method (BWM); goal programming (GP); fuzzy logic; electronics industry

## 1. Introduction

The supply chain encompasses all sections contributing customers' orders [1]. In modern supply chain, the natural or man-made disruptions can result in a disturbance in raw material supply flow, which can affect business flow. The current business environment causes high levels of uncertainty and chaotic behaviors in supply chains [2]. These chaotic behaviors are provoked by factors such as globalization, enhancing the level of outsourcing activities, increasing the demand turbulence, decreasing the life cycle of products, a sharp decline in inventories, and decreasing the number of suppliers [3].

Moreover, the supply chains face great challenges and threats such as: natural disasters (flood, earthquake, typhoons, and conflagration), cyber-attacks, sanctions, supply chain disruptions, production, distribution, etc. In general, the supply chains that are exposed to disruptions and their competitiveness does not rely solely on decreasing costs, higher quality, decreasing the delivery time, or higher levels for customer service. Rather, it can be dependent on their capability in inhibiting and overcoming various disruptions endangering their performance. Therefore, it is important for supply chains to be resilient [4].

Christopher and Peck (2004) propose that the supply chains risks resources are categorized into five levels including the processes and value chain risks, control-related risks, supply risks, demand risks, and environmental risks [5]. In another categorization, these risks are considered as internal risks (process), network risks (supply and distribution), and external risks (environmental) [6].

Based on the supply chain risk resources, the disruptions can enter supply chains in both internal and external forms [7]. The suppliers are among the main factors of external risk and can create broad disruptions in supply chains [8], since the raw material expenses as the main production expense encompass more than 70 percent of the production costs [9]. One of the most important goals in supplier selection is related to the supplier role in meeting a company's needs with appropriate costs. On the other hand, the producers try to

meet their needs through transactions with suppliers in order to increase productivity and create higher levels of value for their customers by considering their purchasing power. The main role of suppliers is rooted in their impact on the process of an organization growth, and this growth is achieved when organizations establish an inseparable and appropriate relationship with suppliers. Organizations are aware of their decisions about choosing suppliers based on the fact that they can be important partners in supply chain. Several studies have focused on supply chain risks and vulnerability, and some studies have noted vulnerability reduction [2,10–13].

However, there are few investigations on supply chain resilience and influential factors in supply chain. The concept of resilience in supply chain was first proposed by Sheffi (2005). He defined resilience as the supply chain conformance capability to decrease sudden disruptions and resistance against the extending of the disruptions through controlling structures and operations, retrieving and reflecting by emergency reactive and effective plans. The supply chain capability in accelerated returning to sustainable operational condition can also be referred to, which can positively influence business performance [14].

Thus, one of the most influential approaches in supply chain management is the resilience approach, which aims to increase the supply chain capability and flexibility in timely responses to customers' demand changes [15]. Supply chain resilience is doubly important due to globalizing the selection process of supply chains and the effects that this process can take from the political and cultural aspects besides the lack of appropriate infrastructures and technologies and the issue of transportation and production [16]. Achieving competitive advantage and improving the supply chain performance is one the main reasons for applying a resilience concept in studying supply chains. Organizations can benefit from a reduction in cost, efficiency improvement, productive usage of resources, and appropriate response to customers' needs by creating resilience capability in their supply chain [17]. Based on the appropriate selection of suppliers in supply chains and regarding the importance of globalization and the competition between various economies, we currently observe supply chains competitions as well.

One of the most influential factors in selecting a successful supply chain is choosing its supplier. Since organizations look for relative superiority in supply chain competitive-ness, the selection of an ideal supply chain can be considered as a great importance. The chain performance improvement in terms of technology, price, quality, and strategic success can be the results of an appropriate supplier selection [18]. Based on the logistics management important role, many companies believe in supplier selection as the most important activity of a business [19]. Thus, inappropriate selection of suppliers can end in irreversible losses for businesses [19]. Moreover, regarding the increasing growth in the outsourcing process on various grounds, appropriate selection of the suppliers plays an important role in the success of organization [13]. According to the abovementioned reasons, selecting an appropriate and resilient supplier can decrease purchase costs and time to a great extent and increase business survival capability in disruptions (such as sanctions, exchange rate turbulences, incomplete industrial infrastructures, changes in demands, customers' needs, fast technological changes, low quality of suppliers products, suppliers activities disorders, and suppliers' inflexibility) and, as a result, increase the company's competitiveness and customer satisfaction [20].

In order to decrease the supply chain risks, it is necessary to have a comprehensive assessment of the resilience concept and provide organizations with these factors in different categories. In fact, organizations will be able to evaluate their resilience as a concept in their risk management in order to plan for its improvement. This paper aims to identify and investigate evaluation and selection factors of resilient suppliers in terms of importance and impact in the electronic industry of Iran.

Reviewing the resilience research background shows that this field is one of the topics that has many applications in the real world and provides the basis for numerous re-search opportunities. Numerous studies have used resilience indicators in evaluating suppliers, including studies by Halder et al. [21,22], Azadeh et al. [23], Rajesh

and Ravi [8], and Saho et al. [24]. However, the applied factors in this research are limited in some aspects. Therefore, the current study has tried to provide a comprehensive review of resilience indicators; so, it can be distinguished from previous research (such as [8,21–24], Jiawu Gan et al. [25], Amindoust. [26], Sureeyatanapas et al. [27], Hasan et al. [28], Davoodabadi et al. [29], and Hasan et al. [30]).

From the other point of view, previous research applied various models to facilitate the supplier selection process. For example, a new hybrid methodology (Panitas et al. (2020)), fuzzy TOPSIS (Hasan et al. (2020)) in the logistics industry, a new interval-valued intuitionistic fuzzy evaluation framework (Davoodabadi et al. (2019)), fuzzy ordinal priority approach (OPA) (Mahmoudi et al. (2021)), data envelopment analysis (Amindoust (2018)), and the BMW and modular TOPSIS in accidental environments (Jiawu Gan et al. (2019)). However, the current study applied hybrid mathematical modeling. An attempt has been made to use the screening method simultaneously, the best–worst method, and goal programming in the fuzzy environment. The proposed hybrid model in this study tries to facilitate the supplier selection process when the supporting information is in-complete; i.e., the uncertain situation for supplier performance assessments that include particular consideration of disruptive situations that have not happened before.

Additionally, the best-known methods from the group of the subjective methods of determining the weighting of criteria are the following: the AHP method, the decision-making trial and evaluation laboratory (DEMATEL) method, the step-wise weight assessment ratio analysis (SWARA) method, and the BWM. Each of these methods has a wide application in the various areas of science and technology, as well as in solving real-life problems. Using such a systematic pairwise comparison (BWM) enhances the consistency and reliability of results. The most important advantage of the BWM is fewer PCs in the traditional AHP process and leads to more consistency compared to other methods.

In this research, the main factors of the suppliers' resilience are identified through a literature review. The factors will be assessed by the experts of the industry and academia by a fuzzy screening questionnaire. At the end, the most important and influential suppliers' resilience factors will be identified by applying decision-making techniques including the best–worst method and goal programming.

## 2. Theoretical Backgrounds

### 2.1. Supply Chain Resilience

There are several definitions of the resilience of supply chain by various researchers in research performed since 2003. The most important definitions are mentioned below. Pregenzer believes that a company's resilience refers to a factor for assessing a company's capability to encounter unexpected and frequent changes in order to protect companies' vital activities [31]. Xiao defines supply chain resilience as the company's capability to return to its primary ideal condition after external environment turbulences. This capability is involved with disruption retrieval and adjustment with the business environment [32]. Hohenstein holds the view that supply chain resilience refers to a chain capability or preparedness to encounter probable and unpredicted hazards, react to the potential deficiencies, and return to the previous state of the business with an appropriate growth through moving toward desirable situation with the aim of customer satisfaction. According to Ponomarov and Holcomb (2009), the supply chain resilience evaluation and measurement can be taken as an important research area for future studies, which will provide appropriate knowledge from this concept [33].

### 2.2. Literature Review

An investigation of the previous research in the supply chain resilience area indicates the fact that most studies that have focused on identifying influential factors in resilience and evaluating these factors and proposing the models for resilience measurement are scarce. Sheffi (2005) defines supply chain resilience as the company's capability to return to its normal performance level in production and service offering after a disruption occurs.

He demonstrated that supply chain resilience plays an important role in material and constant or continuous flow of products and, thus, the company's success and competitiveness [14]. Roberta Pereira et al. (2014) refer to the supply chain resilience as the supply chain capability to show a quick reaction to unexpected challenges in a way that it can reach to its previous performance level or higher [34]. In spite of abundant research on supplier's evaluation in resilient chains and suppliers' selection, there are few studies concerning the selection of suppliers in resilient supply chains [35]. The most prominent studies in this realm are as follows. Haldar et al. (2012) introduced a suppliers' selection method in resilient supply chains applying compound methods based on hierarchical analysis process, TOPSIS, and performance quality function development. They applied two categories of indices including technical indices encompassing supply chain density, supply chain complexity, responsiveness, node sensitivity, and reengineering, and producer indices encompassing buffer capacity, supplier's resources flexibility, and lead times [21]. Savic (2013) has evaluated and selected suppliers under the condition of disruptions in the supply chain and allocated orders to chosen suppliers, applying mixed-integer programming modeling [22]. Halder et al. (2014) proposed a strategic and quantitative approach to select resilient suppliers through fuzzy methods. They applied fuzzy TOPSIS with triangular and trapezoidal fuzzy numbers in their research. They used factors such as quality, products capability, customer satisfaction, and product costs [22]. Azadeh et al. (2014) have suggested an integrated approach for selecting suppliers in green-resilient supply chains. They applied network analysis process compound methods and fuzzy DEMATEL to determine weights and the relationships between indices. The data envelopment analysis was used to rank the suppliers [23]. Kamalahmadi and Melat-Parast (2015) proposed an integrated two-phased mixed-integer programming model for selecting the suppliers, allocating the orders with transportation channel options and contingent planning to decrease the negative effects of disruptions and minimizing the whole costs of the network in a resilient supply chain [15]. Sahu et al. (2016) applied the fuzzy VIKOR method to evaluate and select resilient suppliers in a fuzzy environment. They used two categories of general and resilience factors to evaluate suppliers' resilience [24]. Azvedo et al. (2016) developed a model describing a resourcing strategy, which focuses on the possibility of changing suppliers, strategic inventory, decreasing waiting time, clarity of supply chain, flexible transportation, agility, information sharing, and co-operation [36]. Amindoost (2018) formed a compound framework with a fuzzy deduction system and data envelopment analysis to evaluate the factors of resilient supplier selection and rank suppliers [26]. Parkouhi et al. (2019) applied gray DEMATEL and GSAW to evaluate and select important factors for selecting resilient suppliers. Based on the findings, customization and suppliers' capacity level were known as the most important amplifying and debilitating factors. In the end, the vulnerable, sensitive, unsustainable, and flexible suppliers were identified according to the suppliers' scores in two increasing and decreasing dimensions [20]. John et al. (2019) proposed a compound method of triangular fuzzy numbers, the best–worth method, and modular TOPSIS in random environments in order to develop group decision-making processes for the selection of resilient suppliers. The feasibility and globalization of this method was proved with illustrated examples in the end [25]. Yazdani et al. (2019) investigates an extended version of the combined compromise solution method with grey numbers, named CoCoSo-G for short, to measure the performance of suppliers in a construction company in Madrid [27]. Davoudabadi et al. (2019) introduced a new interval-valued intuitionistic fuzzy evaluation framework for solving resilient supplier selection problems. The criteria weights were determined based on the entropy method. The proposed method was applied to the same case studies used by Sahu et al. [24] for comparison [28]. Panitas et al. (2020) applied a new hybrid methodology, which is able to handle various forms of uncertain and incomplete data and is proposed to facilitate the supplier selection process. The proposed methodology is tested with a case of resilient supplier selection in a company producing computer hardware components. The list of criteria was divided into two groups consisting of resilience capability and general criteria commonly considered

critical for electronic components procurement [29]. Hasan et al. (2020) also employed fuzzy TOPSIS to rank alternative suppliers in resilient supplier selection for the logistics 4.0 industry. The originality of this work lies in the fact that the probability–possibility consistency principle was used to determine triangular fuzzy numbers from large-scale temporal data; goal programming was used to determine optimal order allocation for each supplier [30]. Tirkolaee et al. (2020) applied a novel hybrid approach based on the fuzzy analytic network process (FANP) method for ranking criteria and sub-criteria, the fuzzy DEMATEL for identification of the relationships among the main criteria, and the fuzzy TOPSIS for prioritizing the suppliers to enhance the reliability and sustainability of three levels of the supply chain, i.e., suppliers, central warehouses, and wholesalers according to supplier selection problem [37]. In Pamucar et al.'s (2020) research, the relative weights of sustainable supply chain management practices are extracted by the fuzzy best–worst method (F-BWM), which is capable of better modeling human thinking. Afterwards, the traditional combined compromise solution (CoCoSo) method is enhanced by the integration of the normalized weighted geo-metric Bonferroni functions to select the most proper supplier in a supply chain [38]. Mahmoudi et al. (2021) develop an innovative decision-making technology to handle the supplier selection problems arising from the frequent impreciseness and incompleteness in the present day SC reports within the framework of the resilient supply chain management. This study deploys a two-fold decomposition of the core algorithm of the ordinal priority approach (OPA), one for attributes and the other for alternatives. It extends the OPA to the fuzzy OPA (OPA-F) for solving the supplier selection problems. The study illustrates how green and resilience aspects of the SC can be integrated to a better understanding of the resilient suppliers in the wake of the SC disruptions [39]. Torkayesh et al. (2021) put forward a two-phase sustainable multi-tier supplier selection model for food supply chains based on an integrated decision analysis under multi-criteria perspectives with the model estimates supplier selection criteria weights using a combined version of step-wise weight assessment ratio analysis (SWARA) and level-based weight assessment (LBWA) in conjunction with D-numbers [40].

In order to identify and confirm suppliers' resilience evaluation indices, 27 indices were extracted from literature which is shown in Table 1.

**Table 1.** Suppliers' resilience evaluation indices.

| Indices | Definition | Resources |
|---|---|---|
| Observing ability | The capability of observing the whole chain in order to identify potential threats and react to a disruption. | [15,24,41–48] |
| Cooperation | The capability of effectively working with other entities in the supply chain in order to gain mutual benefits such as information and resource sharing to reduce the level of vulnerability. | [14,15,42–44,47–49] |
| Flexibility | The capability of the company and the supply chain in adjusting with changes in a short time and the flexibility and endeavors of suppliers, production system, distribution channels, transportation methods, and multi-skill personnel. | [4,41–43,46–48] |
| Agility | The capability of swift responding to unpredicted changes in demand and supply. | [15,16,42,46,47] |
| Pace | The flexible compatibility pace, which defines the essential time for a recovery from a disruption in the supply chain. | [15,48] |
| Vulnerability | The supplier invulnerability against different hazards and its resilient sales knowledge and operation planning in order to identify and react to various vulnerability resources. | [8,50,51] |
| Research and Development | Having a strong resource and development department to adjust with chaotic changes and create or sustain innovations inside. | [8,52–55] |
| Risks Awareness | The necessity of the suppliers' awareness about the risks related to the assets, organization, and environment to react quickly and increase the resilience capability. | [8,10,11,56–58] |
| Technological Capabilities | The suppliers' capability of technological adjustment with innovations, production progressive technologies, and their processes, which makes them resilient to encounter chaos and technological turbulences. | [8,59–61] |

**Table 1.** *Cont.*

| Indices | Definition | Resources |
|---|---|---|
| Risk Management Culture | Making insure that the suppliers have accepted risk management and have internalized it as a culture. | [42,44–48] |
| Safety | Providing a healthy and safe working environment for employees to prevent impairments and injuries while carrying out operations. | [8,17,61,62] |
| Supply Chain Structural Status | Designing and constructing a network, which facilitates resilience for instance a balance between efficiency, redundancy, and vulnerability. | [12,14] |
| Compatibility and Adjustment Capability | The compatibility dynamic nature of the supply chains makes them capable of recovering from disruptions and returning to the primary or better situation in supply chain operations. | [63] |
| Trust | Trust is a prerequisite for risk sharing among the chain members. The supply chain management is formed based on the trust, which nourishes co-operations, decreases task conflicts, and strengthens decision-making capability in ambiguity and uncertainty. | [16,43,45] |
| Risk and Income Sharing | Risk and income sharing for long-term focus and cooperation between chain partners is important. A chain performs well when all incentives (safety, hazards, costs, and operations bonuses) are shared between members equally. | [8] |
| Sustainability | Sustainability plays an important role in chain resilience. It enables companies to consider partners' policies and activities about ethical and environmental issues in order to decrease the whole network risks. | [44,64,65] |
| Financial Power | Financial power is one of the most important indices, which guarantee the company's survival in the business turbulent environment. The companies cannot continue their operation without profitability. Thus, this index is one of the most important factors in resilience, which affects supply and logistics activities. | [42,48] |
| Knowledge Management Systems | Creating and developing the knowledge and understanding physical and informational structures of the supply chain. | [14,41,43,45,48,54] |
| Information Sharing | Information sharing among chain members decreases risks and minimizes the outcomes of phenomena such as the Bullwhip effect. | [4,15,16,43–48] |
| Redundancy | Policies such as selection of multiple suppliers, investment in surplus, and strategic inventory reserves for encountering disruptions. | [4,15,42,44,48] |
| Complexity | The supply chain complexity is directly related to the nodes and the relationship between them, which may make the chain inflexible and inefficient and increase redundancy. | [4,41–43,45–48,66] |
| Lead Time | The lead time is referred to the time between the order time and delivery time. Longer delivery time creates critical paths in supply network and ultimately increases the chain vulnerability against disruptions. | [4,46–48] |
| Chain Members Distance | Long distances between the company and the suppliers increase the risk of disruptions. | [46,47] |
| Contingent Planning | Predicting the potential events and defining the methods to face them before happening. | [42,45,67] |
| Demand Management Systems | Decreasing the effects of disruptions related to a customer's choice through strategies such as dynamic pricing, etc. | [47] |
| Human Resource Management | Educating personnel to face dangerous events and creating multi task groups. | [11,33,45] |
| Defining a purposive system for evaluating suppliers' performance (suppliers' performance management system) | Applying factors to evaluate and select suppliers, which can decrease disruptions and their effects (such as financial and political constancy, reliability, accountability, and so on) | [47] |

Also in this section applications of BWM in supplier selection, GP in supplier selection, and resilient supplier selection is shown in Table 2.

**Table 2.** Applications of BWM in supplier selection, applications of GP in supplier selection, and resilient supplier selection.

| Section | Title | Writers | Year of Publication |
|---|---|---|---|
| Applications of BWM in supplier selection | A Fuzzy BWM Method for Evaluating Supplier Selection Factors in a SME Paper Manufacturer | Kurniawan and Puspitasari | 2021 |
| | Sustainable supplier selection: A novel integrated fuzzy best–worst method (F-BWM) and fuzzy CoCoSo with Bonferroni (CoCoSo'B) multi-criteria model | Fatih Ecer and Dragan Pamucar | 2020 |
| | Application of improved best–worst method (BWM) in real-world problems | Pamucar et al. | 2020 |
| | Presenting an integrated BWM-VIKOR-based approach for selecting suppliers of raw materials in the supply chain with emphasis on agility and flexibility criteria (Case study: Saipa corporation) | Azizi et al. | 2019 |
| | Resilient Supplier Selection Based on Fuzzy BWM and GMo-RTOPSIS under Supply Chain Environment | Jiawu Gan et al. | 2019 |
| Applications of GP in supplier selection | A GP-AHP approach to Design Responsive Supply Chains for Pareto Customers | Reza Khorramshahgol and Raed Al-Husain | 2021 |
| Resilient supplier selection | Resilient Supplier Selection in Electronic Components Procurement: An Integration of Evidence Theory and Rule-Based Transformation into TOPSIS to Tackle Uncertain and Incomplete Information | Panitas Sureeyatanapas et al. | 2020 |
| | Resilient supplier selection to mitigate uncertainty: soft-computing approach | Dipika Pramanik et al. | 2020 |
| | Resilient supplier selection in complex products and their subsystem supply chains under uncertainty and risk disruption: A case study for satellite components | Gheidar-Kheljani | 2019 |
| | Resilient supplier selection and optimal order allocation under disruption risks | Seyedmohsen Hosseini et al. | 2019 |
| | Resilient Supplier Selection Based on Fuzzy BWM and GMo-RTOPSIS under Supply Chain Environment | Shuqi Zhong et al. | 2019 |
| | Resilient Supplier Selection Based on Fuzzy BWM and GMo-RTOPSIS under Supply Chain Environment | Shuqi Zhong et al. | 2019 |

## 3. Research Methodology

This research is applied research in terms of its aim and is a descriptive survey in terms of data gathering, since it identifies and describes suppliers' resilience evaluation indices in the electronics industry. A field study method was applied to distribute questionnaires among experts to investigate the importance of these indices.

The purposive sampling method was used to select experts and specialists, since their views were directly related to the research results, and experts' selection is one of the most important stages of the current study. The experts' team encompassed 10 members, including 5 experienced specialists in the electronics industry, and 5 university professors working on operational and research projects in the field of supply chains of the country.

The nature of the comparison process is uncertain and vague. This feature causes the decision maker not to be able to make his/her preferences clear. That is to say, human judgments are usually accompanied by ambiguous preferences. Crisp values could not reflect the decision makers' vague opinions. So, applying the precise pair-wise comparison is debatable. Using linguistic evaluations instead of numerical values could be a realistic approach. The theory of fuzzy systems can enter human knowledge, experience, judgment, and decision in the model, by applying fuzzy logic theory and fuzzy measure theory. It makes the results of such models more practical and accurate. Concerns about an absolute scale, with no degrees of freedom, caused a tendency toward the use of a fuzzy modification of the MADM in different fields of research [68]. Given the advantages of fuzzy set theory and fuzzy MADM reported by different researchers, especially for SCM issues, this study applied it for a more accurate extraction of the experts' opinions [69].

Aspects of works on BWM have been applied in several hybrid approaches for providing decision-making models in different fields, such as suppliers evaluation and selection [66,70]; providing an enhanced risk assessment method within the business continuity management system framework [37] (Torabi et al., 2016); selecting the best enablers of technological innovation in India [38]; selecting the most proper type of bundling configuration in the surface transportation [25]; assessing the importance of different types of energy such as oil and gas industries on sustainable SCM [71,72]; selecting the proper technology for the treatment of urban sewage [59]; measuring the performance of R&D [60]; designing a SERVQUAL model to evaluate the service quality of a baggage handling system [66].

Aspects of extensions of BWM are applied in uncertain decision environments; BWM has experienced some modifications and extensions. Rezaei (2016) presented a linear model by making some changes in the original BWM steps [66]; Guo and Zhao (2017) and Hafezalkotob (2017) have presented the fuzzy BWM (FBWM) by applying the fuzzy approach and triangular fuzzy numbers [73,74]. Mou et al. (2016) applied intuitionistic fuzzy multiplicative preference relations for ranking criteria or alternatives, and it can be considered as a tool for combining with other MCDM methods [75]. Mou et al. (2017) presented a graph-based group decision-making approach for intuitionistic fuzzy BWM [76]. For considering the uncertainty of the input data in MCDM problems, Aboutorab et al. (2018) have proposed the ZBWM by utilizing the Z-number approach, which has a lower inconsistency ratio compared with the BWM [77]. Mi and Liao (2020) enabled BWM to accept hesitant numbers as input [78]. Hafezalkotob et al. (2019) proposed the interval MULTIMOORA and group interval BWM [79].

## 4. Research Phases

### 4.1. Fuzzy Screening

This method is useful when a small subset must be selected from many options for further consideration. Yager (1993) proposed a fuzzy screening system, which can bring about consensus by considering the least information about factors [80].

The execution of this technique only needs preferred linguistic information with ordinal scale. This enables the experts to provide their knowledge and information about their satisfaction from factors and decision options in the form of linguistic variables such as extremely important to completely insignificant. The ability of working on inaccurate

linguistic preferences helps them use sources with minimal information related to the subject under review.

In other words, each decision maker states their idea about importance level and supply degree of each factor (Table 3).

**Table 3.** Qualitative–linguistic space for evaluating factors and determining their importance.

| Linguistic Words | Defined Symbol | Linguistic Measure |
|---|---|---|
| Extremely important | $S_7$ | OU |
| Very important | $S_6$ | VH |
| important | $S_5$ | H |
| Moderately important | $S_4$ | M |
| Slightly important | $S_3$ | L |
| Low importance | $S_2$ | VL |
| Not important at all | $S_1$ | N |

The fuzzy screening process is a two-phased process [80]:

A.   Information and knowledge gathering from decision-making group members.

In this phase, the decision-making group members are asked to present their judgement about importance level or supply degree of each factor through decision options in the form of defined linguistic words shown in Table 1, which are based on a linear ordinal scale.

B.   Integration and aggregation of decision-making group members' linguistic judgments

In this phase, each member's judgments and fuzzy preferences about each measure's importance or supply degree are integrated and aggregated in order to reach a unique value for each factor. The aggregation function is defined as follows:

$$k = 1.2.3\ldots.r \quad Q_A(k) = S_{b(k)} \qquad b(k) = Int\left[1 + \left(k\frac{q-1}{r}\right)\right] \tag{1}$$

In this equation, q is the number of points in selected scale; $r$ is the number of experts involved in decision-making process; *Int* is the measure of the integer; $k$ is the number of experts supporting the alternative.

After an appropriate consensus function, the ordered weighted average (OWA) function can be used to aggregate decision-making group members' (experts) ideas. The OWA is an effective method to aggregate individuals' linguistic preferences in a group collective preference.

The first step in this phase is an aggregation function such as $Q$ for decision-making body. This function demonstrates the agreement of definite numbers of decision-making team members on each factor's importance or supply degree based on decision alternatives and that factor screening as a key factor or selection of that alternative as the most appropriate alternative.

The value $Q$ (k) shows if the kth member, diagnoses factor I as a key factor first, and then, chooses that factor as the best one and how that factor will be selected.

*4.2. The Best–Worst Method*

In multiple-choice decision-making methods, some alternatives are evaluated based on some factors in order to choose the best alternative. Based on the best–worst method proposed by Rezayi (2015) [66], the best and worst factor is identified by the decision maker, and paired comparisons between these two factors (the best and the worst) and other factors will be carried out. A maximum–minimum problem will be formulated and solved afterwards to determine different factors weights. In this method, a formula to calculate the inconsistency rate is considered to investigate the reliability of paired comparisons.

This method has been used in many studies based on its applicability [70,81,82]. One of the most eminent features of this method is that it needs less comparison data compared

to other multiple criteria decision-making methods and results in more robust comparisons and more reliable conclusions. There are several extended versions of BWM. Most of the recent BWM/F-BWM contributions focus on supply chain design, supplier or green supplier evaluation, and occupational or environmental safety risk analysis. For supplier development problems, intuitionistic F-BWM is applied for the green supplier selection problem. The Bayesian BWM is based on the original BWM, so the input, i.e., the pairwise comparisons, is the same. However, as for the output, there is a difference between the two methods. In the original BWM, the final output is a concrete value of the weight, while the Bayesian BWM provides a probability distribution.

The operational stages of the research are as follows:

Step 1   Determining a set of decision-making criteria. In this step, a set of criteria needed for decision making is defined as $\{C_1, C_2, \ldots, C_n\}$.

Step 2   Determining the best (the most important or the most desirable) and the worst (the least important or the least desirable) criteria. In this step, the decision maker determines the best and the worst criterion without comparisons yet.

Step 3   Determining the preference of the best criterion to other criteria using numbers 1 to 9. The preference vector of the best criterion to other criteria is demonstrated as $A_B = (a_{B1}, a_{B2}, \ldots, a_{Bn})$.

In the mentioned vector, $a_{Bj}$ shows the preference of the best criterion (B) to the criterion (j) and $a_{BB} = 1$.

Step 4   Determining the preference of all criteria to the worst criterion using numbers 1 to 9. The preference vector of other criteria to the worst criterion is demonstrated as $A_w = (a_{1w}, a_{2w}, \ldots, a_{3w})^T$.

In the mentioned vector $a_{jw}$, the mentioned vector $a_{jw}$ shows the preference of criterion (j) to the worst criterion (w) and $a_{ww} = 1$.

Step 5   Exploring the optimum measures of weights $(W_1^*, W_2^*, \ldots, W_n^*)$. In order to determine the optimum weight of each criterion, the pairs $\frac{W_j}{W_w} = a_{jw}$, $\frac{W_B}{W_j} = a_{Bj}$ will be formed, and then, to create all conditions in all js, a solution must be found to maximize $\frac{W_B}{W_j} - a_{Bj}$ and for all minimized js. Based on non $\frac{W_j}{W_w} - a_{jw}$ of the weights and the sum of the weights, Equation (2) is formulated as follows: Equation (2) is formulated as follows:

$$
\begin{aligned}
\min max_j &\left\{ \left| \frac{W_B}{W_j} - a_{Bj} \right|, \left| \frac{W_j}{W_w} - a_{jw} \right| \right\} \\
&s.t. \\
&\sum_j W_j = 1 \\
&W_j \geq 0, for\ all\ j
\end{aligned}
\tag{2}
$$

It is also possible to transform the model above to the model below:

$$
\begin{aligned}
\min &\ \xi \\
&s.t. \\
\left| \frac{W_B}{W_j} - a_{Bj} \right| &\leq \xi, for\ all\ j \\
\left| \frac{W_j}{W_w} - a_{jw} \right| &\leq \xi, for\ all\ j \\
\sum_j W_j &= 1 \\
w_j &\geq 0, for\ all\ j
\end{aligned}
\tag{3}
$$

The linear model of the function above is shown below. In this study, the criteria weights are calculated applying the linear form of the model.

$$\min \xi$$
$$s.t.$$
$$\left| W_B - a_{Bj} W_j \right| \leq \xi, for \ all \ j$$
$$\left| W_j - a_{jw} W_w \right| \leq \xi, for \ all \ j \tag{4}$$
$$\sum_j W_j = 1$$
$$w_j \geq 0, for \ all \ j$$

After solving the model above, the optimum measures of $(W_1^*, W_2^*, \ldots, W_n^*)$ and $\xi^*$ will be calculated. The greater measure of $\xi^*$ is demonstrative of higher consistency rate. Consistency rate can be produced using a consistency index (Table 4 and Equation (5)). The closer consistency measure to zero is demonstrative of higher consistency results.

$$Consistency \ Rate = \frac{\xi^*}{Consistency \ index} \tag{5}$$

**Table 4.** Consistency criteria applying BWM method.

| $a_{BW}$ | 9 | 8 | 7 | 6 | 5 | 4 | 3 | 2 | 1 |
|---|---|---|---|---|---|---|---|---|---|
| Consistency index | 5.23 | 4.47 | 3.73 | 3 | 2.3 | 1.63 | 1 | 0.44 | 0 |

*4.3. Fuzzy Goal Programming*

Goal programming is one of the most robust models in mathematics with great capability of solving multi-purpose problems specially when there are contradictory goals. According to the ambiguity of the information gained from the environment, applying fuzzy goal programming seems to be logical. In this study, Zimmermann's fuzzy goal programming is used. The modeling of this method is as follows:

$$maxz = \sum_{j=1}^{Q} wj\lambda j$$
$$s.t.$$
$$\lambda_j \leq \mu_{zj}(x) \quad j = 1.2 \ldots . q \quad (For \ all \ objective \ functions) \tag{6}$$
$$\gamma_r \leq \mu_{gr}(x) \quad r = 1.2 \ldots . h \quad (For \ fuzzy \ limitations)$$
$$g_p(x) \leq b_p \quad p = h + 1 \ldots . m \quad (For \ certain \ limitations)$$

Membership functions for maximum goals are demonstrated in Equation (7).

$$\mu_{zj}(X) = \begin{cases} 1 & zj \geq zj^+ \\ \frac{zj(x) - zj^-}{zj^+ - zj^-} & zj^- \leq zj(x) \leq zj^+ \\ 0 & zj \leq zj^- \end{cases} \tag{7}$$

Membership functions for minimum goals are demonstrated in Equation (8).

$$\mu_{zj}(X) = \begin{cases} 1 & zj \leq zj^- \\ \frac{zj^+ - zj(X)}{zj^+ - zj^-} & zj^- \leq zj(x) \leq zj^+ \\ 0 & zj \geq zj^+ \end{cases} \tag{8}$$

Membership functions for fuzzy limitations are demonstrated in Equation (9).

$$\mu_{gr}(X) = \begin{cases} 1 & g_r(x) \leq b_r \\ 1 - (g_r(x) - br)/dr & br \leq g_r(x) \leq b_r + d_r \\ 0 & g_r(x) \geq b_r + d_r \end{cases} \quad (9)$$

## 5. Research Findings

### 5.1. Confirming Suppliers' Resilience Criteria

In order to confirm suppliers' resilience criteria, 27 criteria extracted from the literature were used in a questionnaire for fuzzy screening, and the experts were asked to answer the questions following this method. According to the prerequisites determined by the experts, if a criterion receives OU score, it will be selected (Table 5).

**Table 5.** Fuzzy screening results.

| Criterion | N | VL | VL | L | M | M | H | H | VH | OU | $u_i$ | Result |
|---|---|---|---|---|---|---|---|---|---|---|---|---|
| Contingent Planning | L | M | M | H | H | H | H | H | H | H | H | × |
| MIN | N | VL | VL | L | M | M | H | H | H | H | | |
| Complexity | M | M | H | H | H | H | VH | VH | VH | VH | VH | × |
| MIN | N | VL | VL | L | M | M | H | H | VH | VH | | |
| Vulnerability | M | H | H | H | H | VH | VH | VH | OU | OU | OU | ✓ |
| MIN | N | VL | VL | L | M | M | H | H | VH | OU | | |
| Knowledge Management | L | M | M | M | M | M | H | H | H | H | H | × |
| MIN | N | VL | VL | L | M | M | H | H | H | H | | |
| Agility | H | VH | VH | VH | OU | OU | OU | OU | OU | OU | OU | ✓ |
| MIN | N | VL | VL | L | M | M | H | H | VH | OU | | |
| Risk Awareness | M | M | M | M | M | M | H | H | H | H | H | × |
| MIN | N | VL | VL | L | M | M | H | H | H | H | | |
| Distance | L | M | M | M | H | H | H | H | H | H | H | × |
| MIN | N | VL | VL | L | M | M | H | H | H | H | | |
| Information Sharing | H | H | H | H | VH | VH | VH | VH | VH | OU | OU | ✓ |
| MIN | N | VL | VL | L | M | M | H | H | VH | OU | | |
| Space | M | M | M | M | H | H | H | H | VH | VH | VH | × |
| MIN | N | VL | VL | L | M | M | H | H | VH | VH | | |
| Redundancy | H | H | H | H | H | VH | VH | VH | OU | OU | OU | ✓ |
| MIN | N | VL | VL | L | M | M | H | H | VH | OU | | |
| Stability | M | H | H | H | H | H | VH | VH | VH | OU | OU | ✓ |
| MIN | N | VL | VL | L | M | M | H | H | VH | OU | | |
| Trust | M | H | H | H | H | VH | VH | VH | OU | OU | OU | ✓ |
| MIN | N | VL | VL | L | M | M | H | H | VH | OU | | |
| Financial Power | M | H | H | H | VH | VH | VH | OU | OU | OU | OU | ✓ |
| MIN | N | VL | VL | L | M | M | H | H | VH | OU | | |
| Supply Chain Structure | M | M | M | H | H | H | VH | VH | VH | VH | VH | × |
| MIN | N | VL | VL | L | M | M | H | H | VH | VH | | |
| Safety | L | L | M | M | M | H | H | H | H | H | H | × |
| MIN | N | VL | VL | L | M | M | H | H | H | H | | |
| Observability | M | M | M | M | M | VH | VH | VH | VH | OU | OU | ✓ |
| MIN | N | VL | VL | L | M | M | H | H | VH | OU | | |
| Supply Management | H | H | VH | VH | VH | VH | VH | VH | OU | OU | OU | ✓ |
| MIN | N | VL | VL | L | M | M | H | H | VH | OU | | |
| Selecting Appropriate Supplier | H | H | H | H | H | VH | VH | VH | VH | VH | VH | × |
| MIN | N | VL | VL | L | M | M | H | H | VH | VH | | |
| Lead Time | M | H | H | H | H | VH | VH | OU | OU | OU | OU | ✓ |
| MIN | N | VL | VL | L | M | M | H | H | VH | OU | | |
| Human Resource Management | H | H | VH | VH | VH | VH | VH | VH | OU | OU | OU | ✓ |
| MIN | N | VL | VL | L | M | M | H | H | VH | OU | | |
| Research and Development | M | M | M | H | H | H | H | H | VH | VH | VH | × |
| MIN | N | VL | VL | L | M | M | H | H | VH | VH | | |

<div align="center">**Table 5.** *Cont.*</div>

| Criterion | N | VL | VL | L | M | M | H | H | VH | OU | $u_i$ | Result |
|---|---|---|---|---|---|---|---|---|---|---|---|---|
| Co-operation | M | H | H | H | VH | VH | VH | VH | OU | OU | OU | ✓ |
| MIN | N | VL | VL | L | M | M | H | H | VH | OU | | |
| Technological Capability | M | M | M | H | H | H | H | H | VH | VH | VH | × |
| MIN | N | VL | VL | L | M | M | H | H | VH | VH | | |
| Consistency and Compatibility Capability | H | H | H | H | H | H | VH | VH | VH | OU | OU | ✓ |
| MIN | N | VL | VL | L | M | M | H | H | VH | OU | | |
| Risk Sharing | M | H | H | H | H | VH | VH | VH | VH | VH | VH | × |
| MIN | N | VL | VL | L | M | M | H | H | VH | VH | | |
| Risk Management Culture | H | H | H | H | H | VH | VH | VH | VH | OU | OU | ✓ |
| MIN | N | VL | VL | L | M | M | H | H | VH | OU | | |
| Flexibility | H | H | VH | VH | VH | VH | OU | OU | OU | OU | OU | ✓ |
| MIN | N | VL | VL | L | M | M | H | H | VH | OU | | |

Having analyzed the fuzzy screening questionnaire's data, 12 criteria were confirmed and selected (Table 6).

<div align="center">**Table 6.** Confirmed criteria.</div>

| $C_n$ | Criteria | $C_n$ | Criteria |
|---|---|---|---|
| C1 | Redundancy | C7 | Flexibility |
| C2 | Consistency and Compatibility Capability | C8 | Agility |
| C3 | Trust | C9 | Risk Management Culture |
| C4 | Vulnerability | C10 | Human Resource Management |
| C5 | Information Sharing | C11 | Supply Management |
| C6 | Observability | C12 | Co-operation |

### 5.2. The Criteria Relative Importance

After identifying important evaluation criteria of resilience, the relative importance of them (weights) must be calculated. Thus, the best–worst method was used as a robust tool for paired comparisons. Based on this method, the best and worst criteria were determined first and then compared to other criteria. In the end, their relative importance will be determined. The calculation stages for one expert are shown below, which is exactly repeated for other experts as well.

In Table 7 after identifying the best and worst criteria, the preferences of the best criterion to other criteria and also the preference of other criteria to the worst criterion are determined by numbers 1 to 9.

<div align="center">**Table 7.** Paired comparisons of the criteria with the best and worst criteria.</div>

| | C1 | C2 | C3 | C4 | C5 | C6 | C7 | C9 | C10 | C11 | C12 |
|---|---|---|---|---|---|---|---|---|---|---|---|
| BEST:C8 | 4 | 2 | 5 | 3 | 4 | 3 | 6 | 7 | 8 | 9 | 5 |
| WORST:C11 | C1 | C2 | C3 | C4 | C5 | C6 | C7 | C9 | C10 | C12 | |
| | 6 | 8 | 5 | 7 | 6 | 7 | 4 | 3 | 2 | 5 | |

In Table 8, the next phase of the best–worst method is shown in which modeling is done based on the preference vectors made of paired comparisons in the previous step. Ultimately, the model is entered to LINGO version18 software, and the criteria weights for each expert are calculated. To determine the final weights of the criteria, the average of set of weights calculated for each expert is produced.

**Table 8.** Modelling and model solution.

| MIN ξ | ξ | 0.052613 | All Experts' Average | |
|---|---|---|---|---|
| \|C8-4*C1 \|≤ ξ, | C1 | 0.073658 | C1 | 0.086152 |
| \|C8-2*C2 \|≤ ξ, | C2 | 0.147317 | C2 | 0.153012 |
| \|C8-5*C3 \|≤ ξ, | C3 | 0.058927 | C3 | 0.063439 |
| \|C8-3*C4 \|≤ ξ, | C4 | 0.098211 | C4 | 0.101609 |
| \|C8-4*C5 \|≤ ξ, | C5 | 0.073658 | C5 | 0.07486 |
| ... | C6 | 0.098211 | C6 | 0.092003 |
| \|C6-7*C11 \|≤ ξ, | C7 | 0.049106 | C7 | 0.047568 |
| \|C7-4*C11 \|≤ ξ, | C8 | 0.24202 | C8 | 0.227191 |
| \|C9-3*C11 \|≤ ξ, | C9 | 0.04209 | C9 | 0.041214 |
| \|C10-2*C11 \|≤ ξ, | C10 | 0.036829 | C10 | 0.035181 |
| \|C12-5*C11 \|≤ ξ, | C11 | 0.021045 | C11 | 0.020132 |
| ∑Cj = 1, Cj ≥ 0 | C12 | 0.058927 | C12 | 0.057639 |

*5.3. Selection of the Resilient Supplier*

In this step, a real problem in the electronics industry is investigated. This active company, based in Shiraz, is one of the specialized companies in the electronics industry in Iran, which operates on the grounds of research, designs, production, and supplying products and services in different realms of electronics technology including radars, electronic wars, aerial electronics and control, weapon electronics, maritime electronics, microwave lamps, measurement equipment, and calibration.

In this research, 20 suppliers of special equipment in one of the most strategic products of the company are investigated. The names of the equipment, product, and suppliers are confidential for security reasons.

5.3.1. Decision Matrix

The information needed about suppliers is gathered through distributing questionnaires among five managers and experts (working in different related departments) in this company. Each supplier is scored 0 to 100 based on the resilience criteria mentioned in the last phase. In the end, the averages of ideas are calculated, and a decision matrix is formed according to Table 9.

**Table 9.** Decision matrix.

| Criteria | C1 | C 2 | C 3 | C 4 | C 5 | C 6 | C 7 | C 8 | C 9 | C 10 | C 11 | C 12 |
|---|---|---|---|---|---|---|---|---|---|---|---|---|
| Weight | 0.086 | 0.153 | 0.063 | 0.102 | 0.075 | 0.092 | 0.047 | 0.227 | 0.041 | 0.035 | 0.020 | 0.058 |
| Min/Max | max | max | max | min | max | max | max | max | max | Max | Max | Max |
| S1 | 7.4 | 6.6 | 7.6 | 5.2 | 6.4 | 5.8 | 5.4 | 5.4 | 7.4 | 7.4 | 5.8 | 6.6 |
| S2 | 5.2 | 5.2 | 6.2 | 6.4 | 5.2 | 4.4 | 3.6 | 3.6 | 4.8 | 5.4 | 5.2 | 4.8 |
| S3 | 8.2 | 7.6 | 8.2 | 2.6 | 7.6 | 7.2 | 7.2 | 7.8 | 8.2 | 8.2 | 7.4 | 8.4 |
| S4 | 3.4 | 2.2 | 2.2 | 9.6 | 2.4 | 1.8 | 1.4 | 2.2 | 2.2 | 2.2 | 2.2 | 1.6 |
| S5 | 9.2 | 9.2 | 9.4 | 2.2 | 8.6 | 8.8 | 9.8 | 9.8 | 9.2 | 9.4 | 9.6 | 8.4 |
| S6 | 6.6 | 4.6 | 5.6 | 7.2 | 4.4 | 4.8 | 3.4 | 2.8 | 3.6 | 4.4 | 4.6 | 4.2 |
| S7 | 3.8 | 2.6 | 3.6 | 9.2 | 3.2 | 2.8 | 2.8 | 2.4 | 2.2 | 2.6 | 3.2 | 3.4 |
| S8 | 8.4 | 9.4 | 8.2 | 3.2 | 8.2 | 7.6 | 8.6 | 8.2 | 9.6 | 9.2 | 8.2 | 9.4 |
| S9 | 2.6 | 1.8 | 1.8 | 9.4 | 3.2 | 1.6 | 2.4 | 1.6 | 2.4 | 2.4 | 1.8 | 2.6 |
| S10 | 6.4 | 4.4 | 5.8 | 7.4 | 4.8 | 4.2 | 3.2 | 3.2 | 5.2 | 4.8 | 4.6 | 3.8 |
| S11 | 7.2 | 8.2 | 7.8 | 3.6 | 6.6 | 6.6 | 7.8 | 7.6 | 7.2 | 7.2 | 6.4 | 7.6 |
| S12 | 8.8 | 8.6 | 8.8 | 2.8 | 9.2 | 8.2 | 8.2 | 8.6 | 8.4 | 8.2 | 8.2 | 8.4 |
| S13 | 6.2 | 5.8 | 6.6 | 5.6 | 7.2 | 5.6 | 5.4 | 4.8 | 6.6 | 6.6 | 5.4 | 5.4 |
| S14 | 6.4 | 7.2 | 7.2 | 4.8 | 5.2 | 7.4 | 6.8 | 6.6 | 7.4 | 7.6 | 6.6 | 6.2 |
| S15 | 3.2 | 2.4 | 2.4 | 8.8 | 2.6 | 2.2 | 2.4 | 2.8 | 1.8 | 3.2 | 2.2 | 2.8 |
| S16 | 4.4 | 3.4 | 5.2 | 8.2 | 4.6 | 3.2 | 4.2 | 4.2 | 3.8 | 4.4 | 3.4 | 3.2 |
| S17 | 9.6 | 9.8 | 9.8 | 1.6 | 8.2 | 8.6 | 9.2 | 9.2 | 8.8 | 8.6 | 9.2 | 8.2 |
| S18 | 5.6 | 5.2 | 6.2 | 6.6 | 6.2 | 5.6 | 4.4 | 4.4 | 5.4 | 5.6 | 4.4 | 4.8 |
| S19 | 5.2 | 3.2 | 4.2 | 8.8 | 4.2 | 2.6 | 3.2 | 3.4 | 2.4 | 3.8 | 2.8 | 2.4 |
| S20 | 8.2 | 8.2 | 8.2 | 3.2 | 9.4 | 8.8 | 8.8 | 9.6 | 8.4 | 8.2 | 8.6 | 8.8 |

5.3.2. Fuzzy Goal Programming

In this phase, the problem is modeled based on the data. In must be noted that according to the experts in all membership functions, the lower limit was taken equal to the equipment weekly demand, and the upper limit was determined according to the need for 500 units. The lower and upper limits show the need for the mentioned equipment that experts have determined based on historical data and current needs. Thus, the membership functions for maximum goals will be similar to Equation (10), and the membership functions for minimum goals will be similar to Equation (11).

$$\mu_{Z1}(X) = \begin{cases} 1 & z_1 \geq 500 \\ \frac{z_1(x)-50}{450} & 50 \leq z_1(x) \leq 500 \\ 0 & z_1 \leq 50 \end{cases} \tag{10}$$

$$\mu_{Z9}(X) = \begin{cases} 1 & z_9 \leq 50 \\ \frac{500-z_9(x)}{450} & 50 \leq z_9(x) \leq 500 \\ 0 & z_9 \geq 500 \end{cases} \tag{11}$$

After determining membership functions, the problem will be formulated as follows. $X_i$ is the decision variable, which is the allocated order to the i th supplier. The coefficients of the objective function are the weights produced applying the best–worst method and show the priority level of each goal. Then, a limitation is written for each goal according to the goal membership function. Thus, there will be 12 goal limitations. Moreover, the suppliers' capacity limitation and demand limitation exist besides goal limitation as well.

$$
\begin{aligned}
&\text{MAX } 0.086\,\lambda_1 + 0.153\,\lambda_2 + \ldots + 0.020\,\lambda_{11} + 0.058\,\lambda_{12} \\
&\text{s.t.} \\
&((7.4\,X_1 + 5.2\,X_2 + 8.2\,X_3 + \ldots + 5.6\,X_{18} + 5.2\,X_{19} + 8.2\,X_{20} - 50)/450) \geq \lambda_1 \\
&((6.6\,X_1 + 5.2\,X_2 + 7.6\,X_3 + \ldots + 5.2\,X_{18} + 3.2\,X_{19} + 8.2\,X_{20} - 50)/450) \geq \lambda_2 \\
&((7.6\,X_1 + 6.2\,X_2 + 8.2\,X_3 + \ldots + 6.2\,X_{18} + 4.2\,X_{19} + 8.2\,X_{20} - 50)/450) \geq \lambda_3 \\
&((500 - 1.6\,X_1 - 2.2\,X_2 - 3.2\,X_3 - \ldots - 8.8\,X_{18} - 9.4\,X_{19} - 9.6\,X_{20})/450) \geq \lambda_4 \\
&((6.4\,X_1 + 5.2\,X_2 + 7.6\,X_3 + \ldots + 6.2\,X_{18} + 4.2\,X_{19} + 9.4\,X_{20} - 50)/450) \geq \lambda_5 \\
&((5.8\,X_1 + 4.4\,X_2 + 7.2\,X_3 + \ldots + 5.6\,X_{18} + 2.6\,X_{19} + 8.8\,X_{20} - 50)/450) \geq \lambda_6 \\
&((5.4\,X_1 + 3.6\,X_2 + 7.2\,X_3 + \ldots + 4.4\,X_{18} + 3.2\,X_{19} + 8.8\,X_{20} - 50)/450) \geq \lambda_7 \\
&((5.4\,X_1 + 3.6\,X_2 + 7.8\,X_3 + \ldots + 4.4\,X_{18} + 3.4\,X_{19} + 9.6\,X_{20} - 50)/450) \geq \lambda_8 \\
&((7.4\,X_1 + 4.8\,X_2 + 8.2\,X_3 + \ldots + 5.4\,X_{18} + 2.4\,X_{19} + 8.4\,X_{20} - 50)/450) \geq \lambda_9 \\
&((7.4\,X_1 + 5.4\,X_2 + 8.2\,X_3 + \ldots + 5.6\,X_{18} + 3.8\,X_{19} + 8.2\,X_{20} - 50)/450) \geq \lambda_{10} \\
&((5.8\,X_1 + 5.2\,X_2 + 7.4\,X3 + \ldots + 4.4\,X_{18} + 2.8\,X_{19} + 8.6\,X_{20} - 50)/450) \geq \lambda_{11} \\
&((6.6\,X_1 + 4.8\,X_2 + 8.4\,X_3 + \ldots + 4.8\,X_{18} + 2.4\,X_{19} + 8.8\,X_{20} - 50)/450) \geq \lambda_{12} \\
&X_{1,3,8,10,13,14,19,20} \leq 5 \text{ and } X_{2,6,7,9,15,16,17,18} \leq 10 \text{ and } X_{4,5,11,12} \leq 15 \\
&X_1 + X_2 + X_3 + \ldots + X_{18} + X_{19} + X_{20} = 50 \\
&X_i \geq 0 \; i = 1, 2, 3, \ldots, 20
\end{aligned} \tag{12}
$$

After modeling the problem, it is solved, and the amount of the order that must be supplied by each supplier is determined. According to the results, the orders allocated to selected suppliers are determined based on their maximum capacity, and no orders will be allocated to other suppliers.

| S5 = 15 | S8 = 5 | S12 = 15 | S17 = 10 | S20 = 5 |
|---|---|---|---|---|

As observed above, the orders allocated to the suppliers 5, 8, 12, 17, and 20 are based on their maximum capacity.

## 6. Discussion and Contributions

Reviewing the related literature shows that several studies have been done to facilitate resilient supplier evaluation. However, their applied criteria are limited in some aspects [8,21–24] (Amindoust [26], Jiawu Gan et al. [25] Sureeyatanapas et al. [27], Hasan

et al. [28], Davoodabadi et al. [29], Mahmoudi et al. [29]). This study tried to fill the research gap in resilient supplier selection in the electronics industry.

Additionally, previous research applied different models for selecting resilient suppliers (Jiawu Gan et al. [25], Sureeyatanapas et al. [27], Hasan et al. [28], Davoodabadi et al. [29], Mahmoudi et al. [30]). These studies have focused on single source procurement in the supply chain, and for this purpose, they have resorted to extracting suppliers' indicators. They evaluated supplier resilience using multi-criteria decision-making methods and introduced the top supplier.

On the other hand, there are studies that have evaluated the resilience of suppliers. They reassigned orders in multiple source sourcing with a mathematical modeling approach. Among this research, we can mention research such as Savik [2], Kamal Ahadi, and nationalist authors [15]. This study aimed to combine the two approaches in this area to have the benefits of both of them. The weakness of comprehensiveness indicators and ignoring them are obvious in the both approaches. Another advantage of the current research is applying fuzzy goal programming, which can overcome the uncertainty of research space with fuzzy calculations.

In research conducted by Jiawu Gan et al. [25], suppliers' selection processes have been described by developing a combinatory method including triangular fuzzy numbers, the best–worst method, and modular TOPSIS in accidental environments. In the mentioned study, the resilient suppliers' selection factors encompass three dimensions and eight factors. These dimensions include absorption capacity (five factors), adaptation capacity (two factors), and renewal capacity (one factor). The distinguishing point between the current study and the previous research is in investigating 27 criteria in different dimensions and applying a combination of different resilient suppliers' evaluation methods.

Amindoust (2018) has also investigated resilient-stable suppliers by considering general factors, resilience, and stability. The indices that he proposed in the resilience factor seem to be a combination of inventory control, managerial criteria, and locating criteria. However, in the current study, these factors have been evaluated in a more comprehensive manner. Another distinction between this study and Amindoust's is in the application of data envelopment analysis to select resilient suppliers.

Sahu et al. [24] has divided resilient supplier factors into two strategies: general strategies (five factors) and resilience strategies (three factors). In this study, product reliability capability has been proposed as a general factor, while in the current research, the compatibility and consistency both demonstrate one single concept. It is also worth mentioning that both responsiveness and agility factors in that research refer to the concept of responsiveness, while agility refers to a quick response to unpredictable changes in demand. The distinctive point between the current study and the research done by Sahu et al. is in the comprehensive investigation of resilient suppliers' selection criteria. Two criteria mentioned in the resilience factor, i.e., investment in capacity buffers and capacity of strategic safety stock, show the researcher's view on inventory control to meet the third factor, responsiveness. However, in the current study, the two mentioned factors are stated in a factor called redundancy, which shows three concepts in one criterion.

Haldar et al. [22] have investigated five factors in the selection of resilient suppliers including quality, product reliability, product performance, customers' satisfaction, and product's cost. Product reliability and product performance in the mentioned research have been considered as components of greater factors called consistency and compatibility capability in this research. Based on the definitions in the current study, the compatibility and consistency capability refer to the dynamic nature of supply chains' consistency capability, which enables them to recover after a disruption and return to their primary situation or a better situation than before in operations of the supply chain. In fact, the product performance and reliability are a consequence of chain recovery after disruptions, which results in customers' satisfaction. An increase in the chain compatibility capability can lead to a better image from product performance and reliability capability.

Pramanik et al. (2016) [83] have investigated the resilient supplier's selection factors in four dimensions, and in resilience dimensions, they refer to five factors including buffer capacity, critical nodes number, responsiveness, reengineering, and compatibility capability. The number of critical nodes is equivalent to the complexity, i.e., the number of nodes and the relationships between them, in this research. This concept has a direct relationship with nodes and relationships between them and can make the chain inflexible and inefficient and increases redundancies. In the mentioned research, this factor has been proposed as the most important factor in the resilience of the supply chain. The compatibility capability definition is different in this research and the mentioned research. In the study done by Pramanik et al. [83], the compatibility capability refers to new knowledge integration and commercialization in order to develop products in the competition process, while in this research, it refers to the chain recovery after disruptions and returning to the normal or better situation. In the end, it is suggested to investigate and extract quantitative evaluation factors of suppliers' resilience in order to decrease the mental judgments of experts. Investigations about the integration of other paradigms with the resilience paradigm in the supply chain are recommended for future studies.

To sum up, our study makes several contributions to the related literature. First, it has provided a review of resilience criteria to construct a comprehensive assessment in the electronics industry. Our list, particularly, the group of resilience capability criteria, can also be used as a starting point for practitioners in other industries, as well as for researchers in future studies, to formulate their own lists for the purpose of resilient supplier selection.

Additionally, the current study applied the fuzzy screening method, BWM, and GP, simultaneously. So, another contribution of this study is the proposed hybrid model to facilitate the supplier selection process when there is uncertainty and incompleteness in supporting information. This issue is considered to be highly probable in assessments of supplier performance that include, in particular, consideration of disruptive situations that have never happened. This study provides a convenient method of aggregation in which various forms of uncertainties and incompleteness in the original assessment results can be used for further analysis. The results obtained are considered to be more realistic and convincing than those provided by other methods constrained by the assumption of a combined precise value (as a consequence of managing uncertainty) or ignoring missing information.

This study enables manufacturers, acting as purchasers, to devise a resilience strategy to minimize the vulnerability of their business processes to any disruptions. The proposed methodology is applicable not only to electronic components procurement, but also to any cases of MCDM applied to other industries, particularly when decision makers perceive insufficient evidence or unavailable information as an obstacle for assessing alternatives.

## 7. Conclusions, Suggestions, and Future Research Directions

According to the increased complexity, uncertainty, and unpredictable changes in global supply chains, the possibility of severe disruptions occurrence in the whole chain has increased, and obviously, the concept of supply chain resilience has attracted more attention than before. So, this study has tried to identify and analyze the indicators of supplier resilience and, ultimately, select the appropriate supplier based on these indicators in Iran's electronics industry. After reviewing the supplier selection indicators and extracting an initial list of the related criteria (including 27 criteria), this research has applied a logical and innovative combination of robust and applicable methods, i.e., fuzzy screening, FBWM, and GP, in the resilient supplier selection model. This model facilitates the process of supplier selection when the supporting information is uncertain and incomplete. By reviewing the literature, 27 criteria are identified for resilient supplier selection (Table 1). Then, the fuzzy screening method is applied to extract a set of more important criteria based on experts' opinions. The findings show that 12 criteria have remained as the most important criteria for resilient suppliers in the electronic industry of Iran. Then, the FBWM is applied to identify the importance of each criterion. The results show that agility (0.227), compatibility

(0.153), and vulnerability (0.102) are the most important factors; other criteria are in the next priorities. Finally, by applying the goal programming (GP), 20 suppliers of special equipment of the company are investigated; five suppliers (S5, S8, S12, S17, and S20) are selected and the orders allocated to them based on their maximum capacity.

The preliminary findings of the study lie in the screening stage and extraction of vital factors of resilient suppliers' selection such as agility, consistency and compatibility capability, and non-vulnerability of the chain.

In the agility dimension in supplier selection, suppliers should pay attention to factors including delivery space and decreasing delays in order to increase competitiveness, since these activities lead to decreased products and services costs. Quick response capability to the market and changes in customers' needs is influential in the selection of agile suppliers.

Compatibility, contrary to risk, refers to the supply chain vulnerability possibility. Several factors, including external demand, supply amount, and even, internal processes, affect suppliers' selection risk in the supply chain. Therefore, it is necessary to devote special attention to concepts, such as inventory management and time in the supply chain, in order to ensure supplier's compatibility and chain disruptions decrease. Thus, it is highly recommended to an organization's managers to select responsive and flexible suppliers and develop their resources and infrastructures in order to decrease the risks of threats in their organizations to a great extent. Compatibility between supplier's performance level and the consumers is possible through more coordination brought by technological tools such as the internet and virtual guidelines.

The suppliers' non-vulnerability the factor is another mentioned criterion in this study. One of the potential sources of vulnerability in the supply chain is the economic turbulences that a supplier might encounter. In fact, organizations must decrease the risks of selecting inappropriate suppliers by identifying suppliers resistant to turbulences with robust economic backgrounds. This research proposes that managers decrease the vulnerability caused by their suppliers by adopting appropriate operational planning and executing various strategies while avoiding disruptions in suppliers' performance such as using alternative suppliers, flexibility in the selection of suppliers' network, and benefiting from outsourcing capacities for operations.

For instance, the resilience paradigm integration with stability, purity, agility, and green paradigms and an integration of research methods are recommended. Besides, it is suggested to apply the decision-making method in the gray environment or hesitant fuzzy method in future studies on account of the qualitative nature of resilient suppliers' evaluation factors and uncertainty of the research space. The evaluation of the suppliers in a resilient supply chain is part of the problem design of the supply chain network. Therefore, it is recommended to study the design of a resilient supply chain network in the electronics industry of Iran applying the proposed model of the current study. Since the electronics industry in Iran is scattered in cities of Tehran, Isfahan, Shiraz, and other cities with production factories, accelerating the production process and increasing the expected quality of products will be possible by designing a network and identifying eminent suppliers.

It should be mentioned that in selecting suppliers and the allocation of orders to the suppliers, only the suppliers meeting the important factors in this research are prioritized. It is suggested that purchasing managers pass educational courses to learn more about supply chains resilience and the procedures through which they can use resilience factors in the decision-making process. It is also suggested that the proposed model in this research be implemented in larger industries to be compared with this research to use the dimensions discussed here in larger industries.

According to the investigations, most managers in this industry are not fully familiar with concepts such as resilience, stability, etc. Furthermore, it is possible to evaluate and rank suppliers periodically in order to improve the suppliers' management. Thus, it is necessary for central offices in Tehran to form teams encompassing purchasing managers of each province to have regular meetings (at least twice a year) in order to identify

eminent suppliers based on the proposed criteria in this research. They can also control the influential activities of suppliers in order to improve the aforementioned factors by periodic evaluations. Due to the importance of the electronics industry in country defense domains and the necessity of decreasing chain disruptions in order to produce world-class products, it is recommended that top managers in the electronics industry supply chains change their viewpoint about suppliers often considered as contractors and take them as their strategic partners based on proposed factors in this research and choose resilient suppliers. It is obvious that changing perspectives in operations and having a close relationship and effective interaction with suppliers will lead to a decrease in disruptions and electronics industry chain risks.

Scientific research always encounters limitations. In this research, the whole community could not be investigated as a result of time, cost, and human resource limitations. Besides, finding knowledgeable experts in the electronics industry with complete awareness about resilient suppliers' selection criteria was considered as a limitation as well.

**Author Contributions:** Conceptualization, H.A. and M.S.N.; methodology, H.A.; software, H.A.; validation, H.A. and M.S.N.; formal analysis, H.A.; investigation, H.A.; resources, H.A.; data curation, H.A. and M.S.N.; writing—original draft preparation, H.A.; writing—review and editing, H.A.; visualization, H.A.; supervision, M.S.N.; project administration, M.S.N.; funding acquisition, H.A. All authors have read and agreed to the published version of the manuscript.

**Funding:** This research received no external funding.

**Conflicts of Interest:** The authors declare no conflict of interest.

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
