# Peer review of "An Integrated Fuzzy Model for Selecting Resilient Suppliers in Electronics Industry of Iran"

_logistics, 2021_

Round 1

Reviewer 1 Report

The paper needs to cite the recent relevant references. In the reference list, the latest references were published in 2019. The sample dataset is too small, which may not lead to a conclusive result. 

Reviewer 2 Report

This paper considers an interesting topic. However, I have the following comments : 

1. The authors should explain the contribution of the proposed model in detail.

2. In the fuzzy model, multiple level of each variables can be considered. The authors just define one level of each variables. Can you please improve the results section in order to show an scenario with multiple level? 

Reviewer 3 Report

This study proposes a MCDM-based tool to address supplier selection in resilient supply chain network for a case study in Iran. Although the paper is addressing an interesting problem, there exist several issues that should be addressed first.

1- Abstract should be modified. Abstract should present the aim of study, methodological contributions and some of the quantitative results. Therefore, try to include you methodology in abstract in details. Give 1-2 quantitative results. Define abbreviation of methods as well.

2- Research gaps should be highlighted first in the introduction. Then, contributions should be highlighted accordingly. In this section, you should also justify why you are using BWM and goal programming. Why not AHP, SWARA, Entropy or CRITIC? Why you are using specifically BWM?

3- Another important thing in the introduction is about justifying why you are using fuzzy logic in order to consider uncertainty. You should justify why fuzzy logic works better than rough numbers, grey numbers, or neutrosophic numbers for you case. For this comparison, you can use information about these uncertainty sets from the following studies:

doi.org/10.1016/j.scs.2021.102712

doi.org/10.1016/j.eswa.2017.08.042

doi.org/10.1016/j.eswa.2021.115354

doi.org/10.1016/j.eswa.2021.115151

4- My biggest concern is about the structure of the literature review which is poor in current format. So my suggestion is: 1- one subsection for applications of BWM in supplier selection in recent years (2019-2021)., 2: one subsection for applications of GP in supplier selection, 3: one section for resilient supplier selection.

My suggestion is to use recently published papers (2019-2021) from the good journals. You can also summarize the studies in each subsection through tables.

5-In the section 3, font size of paragraph 1 and 2 is different.

6- Why FUZZY? you can write fuzzy or Fuzzy. Be consistent. Check for the similar issues in the text.

7- Equations are not aligned in the manuscript. Check it for all of them. You also do not need to bold them.

8- In 4.2., give a short literature review on BWM extensions such as fuzzy BWM, bayesian BWM, or stratified BWM. Then, justify why you selected normal BWM to use rather than other extensions.

9- I suggest to divide conclusion section in two sections of discussions and conclusions. Move the related information to discussions. In current form, it is very messy.

10- The biggest problem in this paper is related to its English language. I strongly suggest authors to ask for a native speaker or an English editor to edit and polish the whole. In current form, it is very hard to follow up.

11- Please check the journal's format and make the modifications. There are some issues in different sections.

12- Add DOI of all references.

Round 2

Reviewer 1 Report

It seems there is no improvement on the dataset. They are still using the same small data.

Author Response

1. It seems there is no improvement on the dataset. They are still using the same small data.
Response: Due to the fact that this article uses a real example, the number of experts has been selected according to the level of access and also the degree of connection with the electronics industry. Also, due to the sensitivity of the studied industry and the confidentiality of information in Iran, there was minimal access to experts aware of the subject of the article.
On the other hand, the data are real and have been obtained based on the type and scope of industry in Iran

Reviewer 2 Report

This paper is ready for publication. 

Author Response

Thank you

Reviewer 3 Report

1- Authors have addressed most of the comments very well. But I think literature review still needs more work. In order to enrich this section, consider the works published by Torkayesh et al., Tirkolaee et al., Yazdani et al., Pamucar et al., and their groups in field of supplier selection in 2020 and 2021.

2- BWM is considered as the most reliable MCDM weighting approach. A few lines for literature review of such as an important in not enough. Consider works on BWM and extensions of BWM such as Stratified BWM and grey BWM.

Author Response

1- Authors have addressed most of the comments very well. But I think literature review still needs more work. In order to enrich this section, consider the works published by Torkayesh et al., Tirkolaee et al., Yazdani et al., Pamucar et al., and their groups in field of supplier selection in 2020 and 2021.
Response: All references mentioned, in section 2-2 have been added in red highlight.
2- BWM is considered as the most reliable MCDM weighting approach. A few lines for literature review of such as an important in not enough. Consider works on BWM and extensions of BWM such as Stratified BWM and grey BWM.
Response: Section 3 Line 266 to 290 in red highlight.
